# SAAMBE-3D: Predicting Effect of Mutations on Protein–Protein Interactions

**DOI:** 10.3390/ijms21072563

**Published:** 2020-04-07

**Authors:** Swagata Pahari, Gen Li, Adithya Krishna Murthy, Siqi Liang, Robert Fragoza, Haiyuan Yu, Emil Alexov

**Affiliations:** 1Department of Physics and Astronomy, Clemson University, Clemson, SC 29634, USA; spahari@clemson.edu (S.P.); genl@g.clemson.edu (G.L.); adithyk@g.clemson.edu (A.K.M.); 2Department of Computational Biology, Cornell University, Ithaca, NY 14850, USA; sl2678@cornell.edu (S.L.); rf362@cornell.edu (R.F.); haiyuan.yu@cornell.edu (H.Y.)

**Keywords:** protein–protein binding, machine learning, stabilizing and destabilizing mutation, disruptive and non-disruptive mutation

## Abstract

Maintaining wild type protein–protein interactions is essential for the normal function of cell and any mutation that alter their characteristics can cause disease. Therefore, the ability to correctly and quickly predict the effect of amino acid mutations is crucial for understanding disease effects and to be able to carry out genome-wide studies. Here, we report a new development of the SAAMBE method, SAAMBE-3D, which is a machine learning-based approach, resulting in accurate predictions and is extremely fast. It achieves the Pearson correlation coefficient ranging from 0.78 to 0.82 depending on the training protocol in benchmarking five-fold validation test against the SKEMPI v2.0 database and outperforms currently existing algorithms on various blind-tests. Furthermore, optimized and tested via five-fold cross-validation on the Cornell University dataset, the SAAMBE-3D achieves AUC of 1.0 and 0.96 on a homo and hereto-dimer test datasets. Another important feature of SAAMBE-3D is that it is very fast, it takes less than a fraction of a second to complete a prediction. SAAMBE-3D is available as a web server and as well as a stand-alone code, the last one being another important feature allowing other researchers to directly download the code and run it on their local computer. Combined all together, SAAMBE-3D is an accurate and fast software applicable for genome-wide studies to assess the effect of amino acid mutations on protein–protein interactions. The webserver and the stand-alone codes (SAAMBE-3D for predicting the change of binding free energy and SAAMBE-3D-DN for predicting if the mutation is disruptive or non-disruptive) are available.

## 1. Introduction

Protein–protein interactions (PPIs) play crucial role in various biological processes including cell regulation and signaling [1,2]. Disruption of PPIs away from native properties could cause diseases [3,4,5,6]. Indeed various studies demonstrated that many disease-associated amino acid variants are located at the protein–protein interfaces, and thus affect PPIs by either altering binding affinity or specificity [7,8,9,10,11]. Knowledge of the impact of the mutation on PPIs is not only important to categorize mutations into pathogenic and benign [12,13], but also to develop therapeutic solutions [14,15] and understand the cause of drug resistance [16,17]. In terms of protein engineering, once the mutation effects are known, one can engineer stable protein having a high affinity towards its partner as required for various applications [18,19]. These limited examples illustrate the importance of assessing the effect of mutations on PPIs. 

Experimentally the effect of mutations on PPIs is investigated either via detection of formation of protein–protein complex or via measuring the change of the binding free energy. In the first case, one categorizes the mutations as disruptive (complex is not formed) and non-disruptive (complex is formed) [20,21]. Such experiments have been done on a large set of mutations providing genome-scale investigations for assessing the effect of mutations on PPIs and results were collected into several databases [22,23,24]. In the second case, one measures the binding affinity of wild type and mutant proteins and then finds the change of the binding free energy (ΔΔG), a procedure which is tedious and time-consuming. Despite of that, over the years many such experiments were reported in literature and collected in various databases as ASEdb [25], PINT [26], AB-bind [27], PROXiMATE [28], DACUM [29], SKEMPI 1.1 [30], and SKEMPI v2.0 [31]. The main distinctions with respect to disease linkage between these two approaches and corresponding databases are the following. Disruptive mutations almost for sure are disease-causing, while non-disruptive may or may not be benign. Thus, a mutation may not completely abolish complex formation, and will be classified as non-disruptive, but may change the binding affinity or specificity to an extent to cause a disease. Regarding approaches and databases inferring the change of the binding free energy caused by a mutation, there is no clear cut-off (magnitude of the binding free energy) that can be used as a marker for identifying the mutation that is disease-causing [12].

On the other side of the spectrum are computational methods to predict the effects of mutations on PPIs [6]. To assess their performance (typically via the Pearson correlation coefficient, PCC) or to train them, the developers use the abovementioned databases. This brings us to an important point that the performance of the computational predictors is limited by the noise and the distribution of experimental data [32]. The wide range of experimental conditions in which experiments are performed to measure ΔΔG increases the uncertainty associated with the measured entries in the database. Therefore, the quality of experimental ΔΔG is important to be considered since it influences the assessment of performance of ΔΔG predictors. In most of the databases, multiple experimental data points (ΔΔGs) using different experimental methods or conditions are reported for the same mutation. Thus, some computational studies used all of these ΔΔGs for training [33], some took the average [34,35,36,37], and some works selected the value coming from the most reliable experimental method [29,38]. It is also important to identify and exclude the mutations with binding affinity outside the experimental detection range, which is not often mentioned in the databases. Therefore, to assure the quality of the experimental data, a number of stringent checks are necessary before using those data as a benchmark for the development of new predictors. Thus, depending on the applied criteria to purge the database(s), one may end up with different set of entries for both the training and test sets, which is one of the main obstacle for direct comparison of predictors performance [39,40].

Various computational methods have been developed to predict the binding affinity changes due to mutation. The earlier methods mostly used physical energy features along with linear regression analysis to predict ΔΔG. Thus, FoldX [41,42] is such a predictor, which uses physical energies such as van der Waals, electrostatic energy, hydrogen bond and solvation energy and allows for conformational changes of the side chains. Another predictor, the Rosetta [43], is based on a linear combination of physical energies such as Lennard-Jones energy, solvation energy and hydrogen bonding. The method also uses the rotamer approach but is restricted to alanine substitutions. In 2009, Concoord/Poisson–Boltzmann surface area (CC/PBSA) method [44] was developed that generates an ensemble of structures and uses the averaged energy of the ensemble to predict ΔΔG. In 2014, another method [45] was developed based on a well-tested simulation protocol, modified MM/PBSA and statistical scoring energy functions and achieved PCC of 0.69 and 0.63 for single and multiple mutations respectively. In addition to physics-based energies, knowledge-based energy terms or statistical potentials were also used to predict ΔΔG. For example, SAAMBE [36,46] uses a combination of MM/PBSA and knowledge-based terms to predict ΔΔG. An important distinction of the SAAMBE method is the usage of amino acid-specific dielectric constant to mimic the conformational flexibility induced by mutation. Another method, the BindProfX [47], combines a conservation profile with the FoldX [41,42] to deliver ΔΔG. One should mention BeAtMuSiC [35], which is a statistical energy-based ΔΔG predictor using a coarse-grained model. All these methods based on either physical energy or statistical potentials or combination of both were demonstrated to achieve PCC ranging from 0.38 to 0.69 on single mutations from SKEMPI 1.1 database.

In recent years, several methods [35,48,49] using machine learning were developed resulting in improved PCC ranging from 0.68 to 0.83. The advantage of machine learning (ML)-based methods is that it can utilize a variety of features based on structure, evolution, energy and others to predict ΔΔG, thus allowing for more extensive “parametrization”. However, the intrinsic data-driven nature of the ML method relies on a large and high-quality experimental dataset. The recent availability of databases containing a large number of experimentally determined ΔΔG catalyzed the development of ML-based method as ΔΔG predictor. The first ML-based predictor is mCSM [33]. It uses the atomic distance pattern surrounding the mutation site to represent the neighboring environment and a regression algorithm is used to train the model. mCSM method was reported to achieve a high PCC of 0.80 on 2317 single mutations from SKEMPI 1.1 database [30]. iSEE [48] is another predictor based on 31 features involving structure interface profile, energy and evolution-based features. It uses a position-specific scoring matrix generated from multiple sequence alignment to describe the evolutionary conservation of the mutation site. iSEE was reported to achieve PCC of 0.8 on single mutations but involving only dimeric complexes from SKEMPI 1.1. MutaBind [37] obtained PCC of 0.68 on the single point mutations of SKEMPI 1.1 with leave one complex out cross-validation, which outperforms FoldX [41,42] and BeAtMuSiC [35] with PCC of 0.40 and 0.39 respectively. Additionally, MutaBind is specially trained on the protease-inhibitor complexes of SKEMPI 1.1, accomplishing PCC of 0.76. Recent development of MutaBind, MutaBind2 [50], resulted in PCC of 0.82 in benchmarking test on SKEMPI v2.0 database. Moreover, MutaBind2 can predict binding free energy change due to multiple mutations. BindProfX [47] uses a structure interface profile to represent the conservation of interface residues. BindProfX combines its estimation of ΔΔG using amino acid probabilities from Boltzmann distribution and FoldX delivered ΔΔG and was shown to achieve PCC of 0.74 on 1131 single mutations from SKEMPI 1.1. However, BindProfX [47] only predicts ΔΔG for mutations located at the interface in the protein–protein complexes. Recently, Rodrigues et al. developed mCSM-PPI2 [49], an improved version of mCSM method. The study demonstrated that mCSM-PPI2 outperforms 26 previous methods on CAPRI blind tests [49]. TopNetTree [51] is another new method that utilized a topology-based network tree by integrating a deep learning algorithm, NetTree, and the topological representation. TopNetTree achieved PCC of 0.82 on SKEMPI v2.0 dataset. The improvement (PCC ranging from 0.75 to 0.83 on SKEMPI v2.0 database, depending on the benchmarking protocol used) is achieved via integrating graph-based signature framework of mCSM method with evolutionary information, inter-residue complex network metrics and energetic terms.

Here we report a new development of SAAMBE [36,46], the SAAMBE-3D, which is a structure-based, ML-based method. The method is very fast and is available as a stand-alone code and as a web server. It utilizes 33 knowledge-based features representing the physical environment surrounding the mutation site. SAAMBE-3D is trained on 3753 single point mutations from 299 complexes from SKEMPI v2.0 [31] using the XGBoost machine learning method and achieves PCC ranging from 0.78 to 0.82 depending on the training protocol. SAAMBE-3D uses a small number of features to avoid the risk of overfitting and it is shown to outperform existing statistical potential and machine learning-based methods on several blind test sets. Furthermore, SAAMBE-3D-DN method is developed to predict disruptive versus non-disruptive mutations. The stand-alone code of SAAMBE-3D, as well as SAAMBE-3D-DN, is available at http://compbio.clemson.edu/saambe_webserver/ and https://github.com/delphi001/SAAMBE-3D.

## 2. Results and Discussion

We trained SAAMBE-3D on a large and diverse dataset containing experimental ΔΔG for 3753 single point mutations from 299 protein–protein complexes. In the dataset, 2892 mutations are at the interface, 2085 are in loops, 1552 are non-alanine mutations, 313 are small to large size amino acid substitutions and 1152 are either small to small or large to large substitutions. For predicting ΔΔG upon a given mutation, we used 33 knowledge-based features only. In order to build a more reliable and robust model, we performed five-fold cross-validation 100 times. Selection of the training and test sets were repeated 100 times randomly, and the average PCC is reported. We trained our model against 80% as well as 90% of the mutation entries present in dataset-1, as described in the method section and tested against the remaining mutations. This way, our model shows a PCC of 0.78 with MSE of 1.23 kcal/mol on 20% of dataset-1 when 80% of dataset-1 is used to train the model (Figure 1a). We were able to obtain a correlation of 0.79 and MSE of 1.25 kcal/mol for 10% of dataset-1 when 90% of the mutations from dataset-1 were used for training the model (Figure 1b). Similarly, for dataset-2, where hypothetical reverse mutations were included to make a more balanced dataset including a similar number of increasing and decreasing binding affinity cases, a correlation of 0.81 and MSE of 1.31 kcal/mol was obtained (Figure 1c). In this case, our model was trained on 80% of the mutations from dataset-2 and tested against the remaining 20% mutations. Using 90% of the mutation data for training the model and 10% for testing, we were able to achieve a correlation of 0.82 and MSE of 1.19 kcal/mol (Figure 1d). A similar performance irrespective of whether we chose 80% or 90% of the mutations from either dataset-1 or dataset-2 to train our model indicates the high stability of our model. To avoid any bias caused by including imaginary reverse mutations, we chose only the model, which is trained against dataset-1 (original data coming from SKEMPI v2.0) for the rest of the validation cases in this paper.

We further evaluated the performance of SAAMBE-3D to assess the quality of classification of mutations into both destabilizing (ΔΔG is positive) and stabilizing mutations (ΔΔG is negative) as well as highly destabilizing (ΔΔG > 1.5) and highly stabilizing mutations (ΔΔG < −1.5). The calculations are done on the dataset-1. First, we identified and put separate labels on the stabilizing and destabilizing or highly stabilizing and destabilizing mutations. We performed receiver operating characteristic (ROC) analysis to quantify the performance of SAAMBE-3D in estimating stabilizing and destabilizing as well as highly stabilizing and highly destabilizing mutations separately. For doing the analysis, we divided the mutations into two classes each time: (i) stabilizing (ΔΔG < 0) and (ii) destabilizing (ΔΔG > 0); (i) highly stabilizing (ΔΔG < −1.5) and (ii) highly destabilizing (ΔΔG > 1.5). Prediction performance is measured by area under the curve, accuracy, precision, sensitivity and Matthews correlation coefficient (MCC) and presented in Table 1. The accuracy is defined as the percentage of correctly categorized mutation (TP and TN) out of the total number of mutations i.e., (TP + TN)/total. Sensitivity is calculated as (TP/TP + FN), specificity is defined as (TN/TN + FP). Additionally, the quality of the prediction is described by MCC in order to account for imbalances in the dataset.
(1)$$\mathrm{MCC}=\frac{\mathrm{TP}\times \mathrm{TN}-\mathrm{FP}\times \mathrm{FN}}{\sqrt{\left(\mathrm{TP}+\mathrm{FP}\right)\left(\mathrm{TP}+\mathrm{FN}\right)\left(\mathrm{TN}+\mathrm{FP}\right)\left(\mathrm{TN}+\mathrm{FN}\right)}}.$$

Figure 2 indicates the excellent performance of SAAMBE-3D in predicting highly stabilizing and highly destabilizing mutations. It achieves MCC of 0.82 and Area Under the Curve (AUC) of 0.99 with an accuracy and precision of 0.96 and 1 respectively. However, the accuracy of SAAMBE-3D in classifying stabilizing and destabilizing mutations is less impressive (Table 1 and Figure 2), achieving MCC of 0.34, AUC of 0.75 with a precision of 0.86 and accuracy of 0.76. The reason is that a significant number of experimental ΔΔG values (1510) of stabilizing and destabilizing mutations are small, in the range of −0.5 to 0.5 kcal/mol, and thus difficult to classify. A ΔΔG of order of ±0.5 kcal/mol is within experimental error.

### 2.1. Further Performance Assessment in Comparison with Existing Methods

All top-performing methods are ML-based method and thus their performance strongly depends on the training and testing sets. For example, we noticed that our old method, SAAMBE, which achieved PCC of 0.62 on the SKEMPI v1.1 database, performed really badly on SKEMPI v2.0, PCC = 0.45. Similar observations were made by other researchers [50]. Therefore, in the next paragraph, we present comparisons and benchmarking for (a) methods that are trained on the same dataset and (b) on blind tests set of data not used in the training.

#### 2.1.1. Comparison of Methods that were Trained on the Same Dataset (SKEMPI v2.0)

We compared the prediction performance of SAAMBE-3D with MCSM-PPI2 [49], and MutaBind2 [50], which are the only two available methods in the literature, trained against SKEMPI v2.0 [1]. We adopted similar purging procedures as the above references, resulting in two datasets: dataset A (which is dataset-1) for comparison with mCSM-PPI2, and dataset B made of 3073 mutations from 257 proteins for comparison with MutaBind2. We believe this is the only way we can make a fair comparison because there is no information available about the training and test sets used for mCSM-PPI2/MutaBind2 methods. The performance of SAAMBE-3D and mCSM-PPI2/MutaBind2 is compared on 3753/3073 single mutation experimental ΔΔG from SKEMPI v2.0 and presented in Figure 3a–c, respectively. The results indicate that SAAMBE-3D (PCC = 0.96 and MSE = 0.23 kcal/mol) outperforms MCSM-PPI2 (PCC = 0.88 and MSE = 0.78 kcal/mol) and MutaBind2 (PCC = 0.89 and MSE = 0.74 kcal/mol) in predicting ΔΔG caused by single mutations.

For further comparison of SAAMBE-3D and mCSM-PPI2/MutaBind2, we considered four different classes of mutations based on the location of mutation site (interface and non-interface), type of secondary structure in which the mutation is located (helix, sheet and loop), type of mutant amino acid (alanine and non-alanine) and change in the size of amino acid side chains (large to large, small to large, large to small and small to small). We compared the predicted values in each of these cases separately with experimental ΔΔG and calculated the PCC using both SAAMBE-3D as well as mCSM-PPI2/MutaBind2 methods. Figure 4 reflects the performance comparison between SAAMBE-3D and mCSM-PPI2/MutaBind2 on mutations present at the interface, non-interface, helix, sheet, loop and involving mutant amino acid residue as alanine or non-alanine separately. Similarly, the performance comparison between these two methods in the form of PCC is presented in Figure 4 for successful prediction of ΔΔG over the mutation involving the different extent of changes in sizes of amino acids.

The results (Figure 4) indicate that SAAMBE-3D performs remarkably well for alanine as well as non-alanine mutations, irrespective of whether mutations are placed at interface or non-interface in the protein–protein complexes, various extent of change in different amino acid side chain sizes. The results show that SAAMBE-3D performs well over wide ranges of mutation types.

Here we continue our comparison between SAAMBE-3D and mCSM-PPI2/MutaBind2 based on complex types. There are four different complex types present in SKEMPI v2.0: (i) protease-inhibitor (ii) antibody antigen (iii) pMHC-TCR and (iv) miscellaneous complexes. (i) The experimental ΔΔG for protease inhibitor complex type ranged from −12.4 to 12.4 kcal/mol. Figure 5 indicates that SAAMBE-3D achieved a correlation of 0.97 with MSE of 0.35 kcal/mol using five-fold cross-validations, whereas mCSM-PPI2 achieved PCC of 0.85 kcal/mol with MSE of 2.07 and MutaBind2 attains PCC = 0.96 and MSE = 0.75 kcal/mol. (ii) pMHC-TCR complexes involve the receptor interacting with various ligands and the experimental ΔΔG for this complex type lies in the range from −2.2 to 7.6 kcal/mol. The PCC of 0.95 with MSE 0.1 kcal/mol were obtained while comparing the SAAMBE-3D predicted data with experimental ΔΔG. On the other hand, PCC of 0.87 and MSE of 0.21 kcal/mol were achieved using the mCSM-PPI2 method.

Here we do not report results of MutaBind2 because the purged database, the database B, has only a few such cases. (iii) Similarly, SAAMBE-3D and mCSM-PPI2 achieve a correlation of 0.95 with MSE 0.28 kcal/mol and PCC of 0.91 with MSE of 0.52 kcal/mol respectively for antibody-antigen complex irrespective of whether mutation present in antibody or antigen. Again, we do not report benchmarking results of MutaBind2 due to limited cases in database B. (iv) The miscellaneous types consist of complexes which do not belong to any of these or any other specific class. For miscellaneous cases, experimental ΔΔG ranges from −5.9 to 8.1 kcal/mol. Figure 5 presents the performance of both SAAMBE-3D and mCSM-PPI2/MutaBind2 on predicting ΔΔG over single mutation for these cases. Figure 5 reveals that for each of these four types of complexes, SAAMBE-3D outperforms mCSM-PPI2/MutaBind2. We can observe in Figure 5 that SAAMBE-3D achieved PCC between 0.95 to 0.97 for all these complex types, which reflects that the method predicts ΔΔG successfully irrespective of complex type. On the other hand, mCSM-PPI2 can predict better for miscellaneous (PCC = 0.92) and antibody-antigen (0.91) compared to protease-inhibitor (PCC = 0.85) and pMHC-TCR complexes (PCC = 0.87). Similarly, MutaBind2 is quite accurate in predicting ΔΔG of protease inhibitor complexes, while less successful in case of miscellaneous.

#### 2.1.2. Performance Comparison on Blind Tests on Set of Data not Used in the Training

Several small datasets available in the literature and not included in SKEMPI v2.0 were frequently used by other researchers for benchmarking. We used these datasets to compare the performance of SAAMBE-3D with other methods (not trained on the corresponding dataset).

First, we used the MDM2-p53 blind datasets presented in Table S8 in the supplementary information of reference [48]. The dataset consists of 33 mutations among which 7 exceeded the experimental detection limit and were removed. Thus, our blind test set consists of 26 mutations from a single protein complex (PDB ID: 1YCR), which was not used for training. We compared PCC of experimental and predicted ΔΔG for available methods. The methods considered for comparison here are iSee [48], mcSM [33], BindProfX [47], FoldX [41], mCSM-PPI2 [49] and MutaBind2 [50]. The PCC values for this dataset using iSee, mCSM, BindProfX and FoldX are taken from Geng et al.’s paper [48]. The results are shown in Figure 6. It can be seen that SAAMBE-3D outperforms all the other methods and achieved PCC of 0.41. Two other predictors with considerable performances are BindProfX (PCC = 0.36) and mCSM-PPI2 (PCC = 0.35).

Another blind set was taken from Benedix el al.’s NM dataset [44]. We only selected single mutations that were not present in our training dataset, further we removed the cases in which mutation site is missing in PDB structure. In this way, 56 single mutations were selected from three complexes (PDB ID: 1IAR, 1VFB and 1YCR). SAAMBE-3D was compared with mCSM-PPI2 [49] and MutaBind2 [50] and the correlation between experimental and predicted ΔΔG for the 56 mutations is presented in Figure 6. We could not calculate ΔΔG using BindProfX as two complexes contain more than two chains and BindProfX can only predict ΔΔG for dimer protein complexes. We also could not make the comparison with iSee, mCSM and FoldX because of the unavailability of the corresponding web server. Figure 6 indicates that SAAMBE-3D is more successful in predicting ΔΔG caused by single mutations compared with mCSM-PPI2. However, MutaBind2 outperforms these two methods including SAAMBE-3D.

The last blind test set was the s487 dataset compiled by Geng et al. [48] The dataset contains 487 mutations from 56 complexes. Geng et al. [48] compared the performance of five different ΔΔG predictors (iSee, BindProfX, FoldX, mCSM and MutaBind2) on these 487 mutations. To compare SAAMBE-3D which was trained on SKEMPI v2.0 database, which in turn includes these mutations, we removed these mutations from our compiled dataset (dataset-1) and retrained our model on 80% of the remaining mutations. Figure 6 represents a comparison of PCCs obtained using different ΔΔG predictors. The PCC values, achieved by iSee, BindProfX, FoldX and mCSM, are taken from Geng at al.’s paper [48] and for MutaBind2, PCC values are obtained from Zhang et al.’s paper [50]. We couldn’t compare the prediction of mCSM-PPI2 as some or all of these 487 mutations are present on their training dataset (SKEMPI v2.0). As shown in Figure 6, BindProfX and MutaBind2 both achieve the highest PCC of 0.41 followed by SAAMBE-3D (0.39). Overall, if we consider the ΔΔG prediction over the three blind datasets, SAAMBE-3D is the most consistent performer as indicated by the PCC values ranging from 0.39 to 0.49. In contrast, some other predictor performance varies significantly depending on the test set. For example, MutaBind2 was the worst performer for the MDM2-p53 test set but the highest performer for NM and s487 dataset.

### 2.2. Further Development of SAAME-3D to Identify Disruptive and Non-Disruptive Mutations both in Homo- and Hetero-Dimeric Protein Complexes (Cornell University Dataset)

We explored how successful our method is in classifying disruptive and non-disruptive mutations in the case of both homo- as well as hetero-dimeric complexes. To detect amino acid mutations that disrupt protein–protein interactions, a high-throughput mutagenesis and cloning platform was previously used to generate clones for each of the mutations reported in our testing set. These mutant clones were then transformed into yeast to perform yeast two-hybrid (Y2H) experiments in which mutations were scored as disruptive if they resulted in significantly reduced Y2H reporter activity relative to corresponding wild-type interactions [52]. A set of 2500 single mutations from 300 homo-dimeric protein complexes and 245 single mutations from 50 hetero-dimeric complexes were measured in Yu lab at Cornell University (termed Cornell University dataset) [53]. The dataset was split into 80% training and 20% test set. We used all the same features for this classification as described in the method section and the resulting code is termed SAAMBE-3D-DN. We performed ROC analysis and found that SAAMBE-3D-DN is 100% successful in classifying disruptive and non- disruptive mutations for homo-dimers (Figure 7). Similarly, we plotted and analyzed ROC for the prediction of disruptive and non-disruptive mutations for hetero-dimer complexes and compared with homo-dimers in Figure 7.

We were able to classify disruptive and non-disruptive mutations in hetero-dimeric protein–protein complexes 96% of the time accurately using SAAMBE-3D-DN. One thing to notice here is that SAAMBE-3D-DN performs well in classifying disruptive and non-disruptive mutations irrespective of whether the complex is homo-dimer or hetero-dimer. Further prediction performance is measured by area under the curve, accuracy, precision and sensitivity, which are presented in Table 2. It has been mentioned in the literature [54] that disruptive mutation is associated with worse outcomes with patients compared to non-disruptive mutations. In this case, SAAMBE-3D-DN is an excellent tool for early diagnostics: consider that one identifies a set of variants in a particular individual and wants to test if these mutations may cause a disease, a disease which is known to be associated with alteration of a set of protein–protein interactions. Then the variants can be subjected to SAAMBE-3D-DN and if predicted to disrupt these interactions, the patient will be diagnosed as having a high risk of developing this particular disease.

### 2.3. Time of Calculations

One of the main considerations in developing SAAMBE-3D method was the prediction should be performed quickly. We determined the time taken for the method execution for each of the entries from dataset-1. The average time is 0.21 secs for one mutation calculation using a single processor. It ranges from 0.18 sec for a small complex (for example 1KNE containing 58 residues) to 0.26 sec for a large complex (4GXU containing 4636 residues). We could not make a fair comparison between SAAMBE-3D and other popular existing methods, because these methods are only available as web servers. However, we compared the time of calculation for a single ΔΔG (C182A) prediction from a given complex (PDB ID: 1A22) using different methods and presented in Table 3 to have a rough estimation of the execution time. One can see from Table 3 that SAAMBE-3D is the fastest method.

## 3. Web Server

We implemented SAAMBE-3D as a user-friendly web server, freely available at http://compbio.clemson.edu/saambe_webserver/. The server is built using JavaScript (Front end) and PHP (Backend). It is hosted on a Linux server running in Apache. SAAMBE-3D can be run for two different kinds of predictions: (i) ΔΔG due to single mutation and (ii) identify whether the given mutation is disruptive or non-disruptive (SAAMBE-3D-DN). Both these predictors can be used in two different ways: (i) predict the effect of mutation specified by the user in the given boxes. The user needs to provide the 3D structure of the protein–protein complex by uploading the coordinate file in PDB format. Users must make sure that the PDB structure file contains at least two protein chains. Users need to provide single mutation information specified by residue number according to PDB file, mutant chain ID, wild type residue in one-letter code and mutant residue in one-letter code. By providing these information in the appropriate box, the user can submit a single job. (ii) If the user wants to submit multiple jobs at the same time, the user can do so by uploading a file called ‘List_Mutations.txt’. The file must contain a list of mutations information in a text file for batch processing. A readme file and a sample ‘List_Mutations.txt’ file are provided in the submission page in order to assist the user for the submission of jobs. Users can also directly download the SAAMBE-3D code by clicking the download option available via the top navigation bar. A readme file will also be downloaded which will guide the user on how to use the code. For each mutation in a given protein–protein complex, the SAAMBE-3D server provides the predicted ΔΔG induced by given mutation. It is important to mention that positive and negative signs in ΔΔG corresponds to destabilizing and stabilizing mutations respectively. For the multiple job option, the predicted ΔΔG are summarized in a downloadable text file in the same order as in the input ‘List_Mutations.txt’ file.

## 4. Methods

### 4.1. Dataset Creation

The dataset used in this work is taken from the SKEMPI v2.0 database [52], which compiles experimental data of binding free energy changes upon mutation in protein–protein complexes. For these complexes, Protein Data Bank (PDB) [55] structures are available. SKEMPI v2.0 contains binding affinity data for both wild type as well as mutant protein–protein complexes for 7085 mutations. These entries include both single point as well as multiple mutations. The database compiled the entries from some earlier databases (ABbind [27], PROXiMATE [28] and dbMPIKT [56]), recent binding affinity data from literature and was manually curated to avoid unreliable entries. Thus, we consider SKEMPI v2.0 to be the gold standard of available experimentally measured binding affinities. In this work, we only considered single point mutations. After filtering the data for only single point mutations, we collected 4169 mutations from 319 different complexes. For some mutations, multiple measurements were reported, and the binding affinity values for all the measurements were listed in the database. If the standard deviation of changes in binding affinity for a given mutation is less than 1.0 kcal·mol^−1^, we considered those cases and used average value for training and benchmarking. We removed all other entries with standard deviation greater than 1 kcal·mol^−1^. As a result, the dataset consists of 4061 single point mutations from 313 different protein–protein complexes. Further, we removed structures, where there are missing residues in the vicinity of the mutation site (five residues in both left and right side from mutation site in the sequence). Thus, the final compiled dataset (dataset-1) is made of 3753 single point mutations from 299 different protein–protein complexes. We observed that among the 3753 entries, for 836 mutations, binding affinity increases going from wild type to mutant, and for 2810 cases, it decreases, and for the rest of the 107 cases, binding affinity is exactly the same in wild type as well as in mutant protein. Following other researchers’ work [49,50], we considered hypothetical reverse mutations. Since the Gibbs free energy change is a thermodynamic state function, change in binding affinity for a given mutation, from wild type to mutant will be equal to the negative change of binding affinity from mutant to wild type. In order to build a balanced set (equal number of cases of positive and negative ΔΔG), we created another dataset (dataset-2), which includes reverse mutations. Thus, dataset-2 contains 7506 single point mutations and represents an increase up to two-folds in datapoints compared to dataset-1. However, we consider this to be an artificial increase of experimental data points, resulting in mirroring the datapoints from the first quadrant to the third one (and vice versa), and providing easier way to obtain better PCC. Because of that, we do not emphasize much on the results using this dataset (dataset-2).

### 4.2. Model Development

Our methodology of predicting binding free energy changes due to mutation in protein complexes, incorporates only knowledge-based features. Overall, we used 33 features (detailed descriptions are provided in the following section), that include the change in molar volume of mutated residue, hydrophobicity, flexibility, hydrogen bond donor/acceptor, polarity, mutation type, chemical nature of the amino acid residue, size of the amino acid due to mutation. The label encoding method is used for incorporating mutation type, change in polarity, chemical properties, hydrogen bond donor/acceptor and size features. We evaluated the performance of all the knowledge-based features separately using the XGBoost machine learning method. In order to avoid overfitting and build a more reliable model, we chose 80% of all mutations as training dataset and used the remaining 20% for testing. To test the stability of the model, we created another model where 90% mutations are selected for training and the remaining 10% are for testing. If the model is not stable, we can expect a large change in PCC between two models (using 80% and 90% as training dataset). For a robust estimate of the mean squared error (MSE) and PCC, this procedure was independently repeated 100 times. For the model building procedure, we performed 5-fold cross-validation to optimize the model parameters. Then, we used the remaining 20% or 10% of mutations as a test set to estimate the performance of the model. For both dataset-1 and dataset-2, we used the same protocol.

### 4.3. Features

#### 4.3.1. Features Related to 3D Structures of the Protein–Protein Complex


The temperature at which the crystal structure is obtained for each complex. This feature accounts for the effect of temperature on the structure as structures obtained at room temperature are typically more flexible (larger atomic B-factors) than structures crystalized at cryogenic temperature.pH of the crystallization for a given protein–protein complex. This feature reflects the fact that ionization states of titratable groups depend on pH, and thus the electrostatic interactions are affected [57,58,59].Resolution of the PDB structures of the complex. The feature account for the structural quality of the corresponding protein–protein complex.


#### 4.3.2. Features Related to Mutation Site


Net volume: The feature represents the change in the molar volume of an amino acid due to mutation. For example, if in a protein–protein complex, a given residue in the wild type is mutated from arginine (R) to alanine (A), then volume change is calculated as molar volume (A)–molar volume (R).Net hydrophobicity: The feature accounts for the change in hydrophobic index (HI) going from wild type to mutant. For example, if in a wild type complex, alanine (A) is mutated to lysine (K) in the mutant, net hydrophobicity will be estimated by HI(K)–HI(A). We considered Moon’s hydrophobic index [60] in this study.Mutation type: We used the label encoding method for this feature. We labeled each different type of mutation. For example, alanine to lysine is labeled 1, lysine to glycine is labeled 2, glutamic acid to arginine is 3 and so on. In this way, there can be a possibility of 380 different types of labels.Net flexibility: The feature estimates flexibility through the presence of a number of rotamers for each residue. Net flexibility represents a change in the number of rotamers going from wild type to mutant residue (see [61] for more details).Chemical property: The label encoding method is applied to seven different chemical properties associated to amino acid residues both for wild type as well as mutant amino acid residue: (i) Aliphatic (A, G, I, L, P, V) (ii) aromatic (F, W, Y) (iii) sulfur (C, M) (iv) hydroxyl (S, T) (v) basic (R, H, K) (vi) acidic (D, E) and (vii) amide (N, Q). Thus, mutations are divided into 49 different categories corresponds to change from wild type to mutant residue through the use of this feature.Size: Amino acids are grouped into three classes according to their sizes: small (G, A, S, C, D, P, N, T), medium (Q, E, H, V) and large (R, I, L, K, M, F, W, Y). The label encoding method is used to create nine different labels representing small to medium, small to small, small to large, medium to medium, medium to large, medium to small, large to small, large to medium and large to a large change in the size of amino acid due to mutation.Polarity: Two polarity classes: polar (R, N, D, Q, E, H, K, S, T, Y) and nonpolar (A, C, G, I, L, M, F, P, W, V) are defined. Therefore 4 different labels are used to represent polar to polar, polar to nonpolar, nonpolar to polar and nonpolar to nonpolar change caused by mutation.Hydrogen bond: Label encoding method uses 16 different labels representing interchange between four classes: (i) donor (R, K, W) (ii) acceptor (D, E) (iii) donor and acceptor (N, Q, H, S, T, Y) and (iv) none (A, C, G, I, L, M, F, P, V) over mutation.Label hydrophobicity: Nine classes are labeled corresponding to interchange between three classes: (i) hydrophobic (A, C, I, L, M, F, W, V) (ii) Neutral (G, H, P, S, T, Y) and (iii) hydrophilic (R, N, D, Q, E, K) due to mutation.


#### 4.3.3. Sequence-Based Features

Along the sequence of the mutated chain, ten residues including five on the left and 5 on the right side from the mutation site are labeled. Each position can be occupied by any of the 20 amino acids, resulting in 20 labels for each position.

#### 4.3.4. Distance-Based Features

We labeled ten nearest residues from the mutation site within 10 Å. These 10 residues belong to chains other than the chain where the mutation site is. This feature captures the effect of inter-chain interaction around the mutation site.

### 4.4. Feature Importance Analysis

We analyzed the importance of each feature selected for the prediction performance of the SAAMBE-3D method. To evaluate the feature importance, we used the algorithm from the python package, XGBoost. Figure 8 represents the contribution of each feature towards the prediction of ΔΔG using SAAMBE-3D. The importance level of each feature is normalized so that the sum of importance level of all the features will be equal to 1. It can be seen in Figure 8 that distance- and sequence-based features are the two most contributing features in our model. The next three important features are the crystallization temperature, change in hydrophobicity and molar volume.

## Figures and Tables

**Figure 1 ijms-21-02563-f001:**
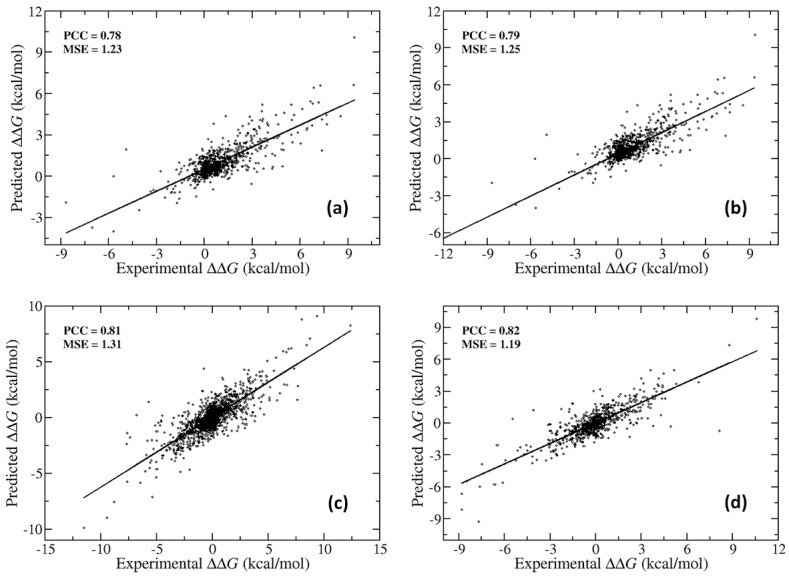
Prediction performance of SAAMBE-3D on dataset-1 for (**a**) 20% mutations as a test set and (**b**) 10% mutations as a test set and on dataset-2 for (**c**) 20% mutations as a test set and (**d**) 10% mutations as a test set.

**Figure 2 ijms-21-02563-f002:**
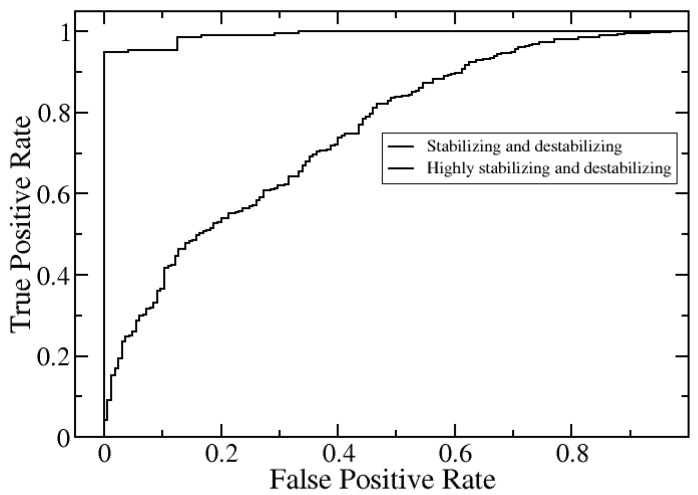
Receiver operating characteristic (ROC) curves for predicting stabilizing/destabilizing and highly stabilizing/highly destabilizing mutations.

**Figure 3 ijms-21-02563-f003:**
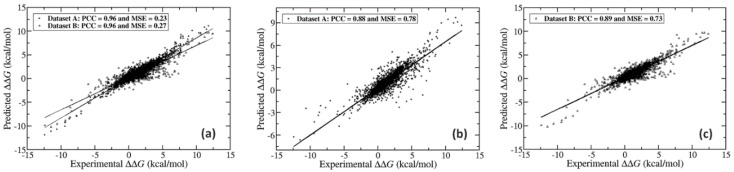
Correlation between predicted and experimental ΔΔG over single mutations on (**a**) both dataset A and B using SAAMBE-3D (**b**) dataset A using mCSM-PPI2 and (**c**) dataset B using MutaBind2.

**Figure 4 ijms-21-02563-f004:**
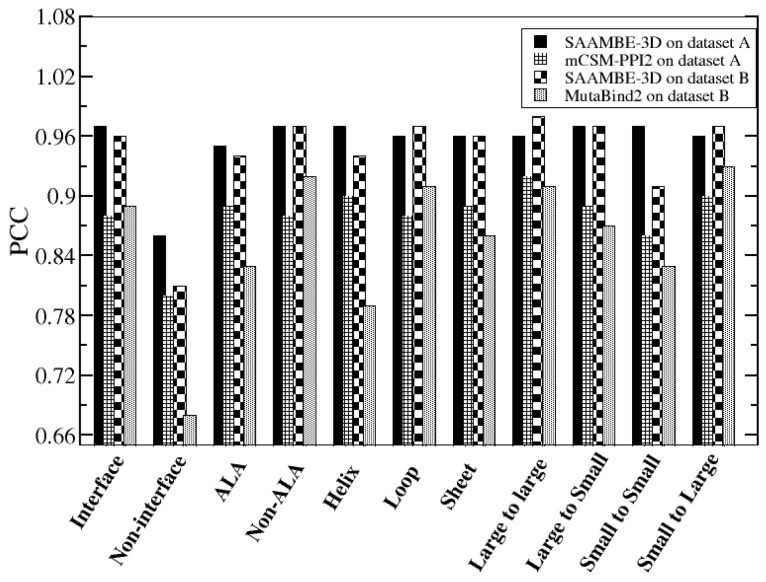
Performance of SAAMBE-3D, mCSM-PPI2 and MutaBind2 on different classes of mutations.

**Figure 5 ijms-21-02563-f005:**
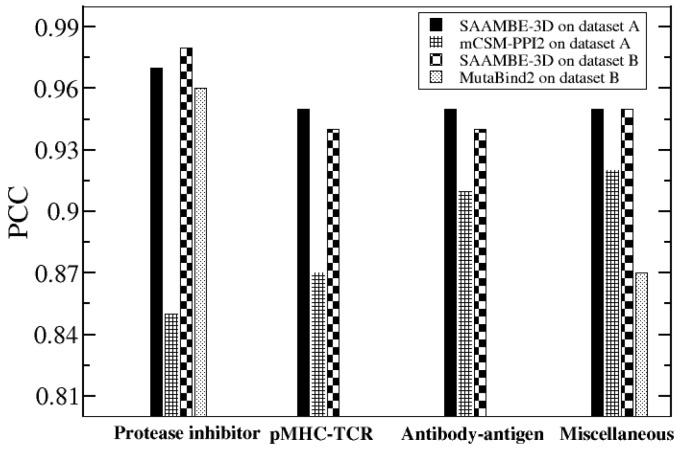
Complex specific performance of SAAMBE-3D, mCSM-PPI2 and MutaBind2.

**Figure 6 ijms-21-02563-f006:**
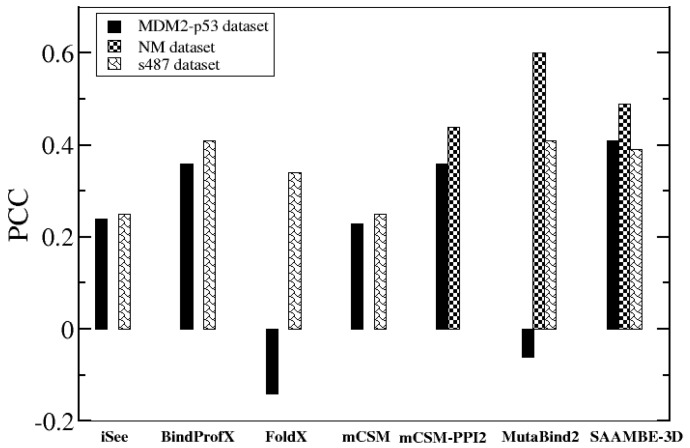
Performance of SAAMBE-3D on three (MDM2-p53, NM dataset and s487 dataset) blind test sets.

**Figure 7 ijms-21-02563-f007:**
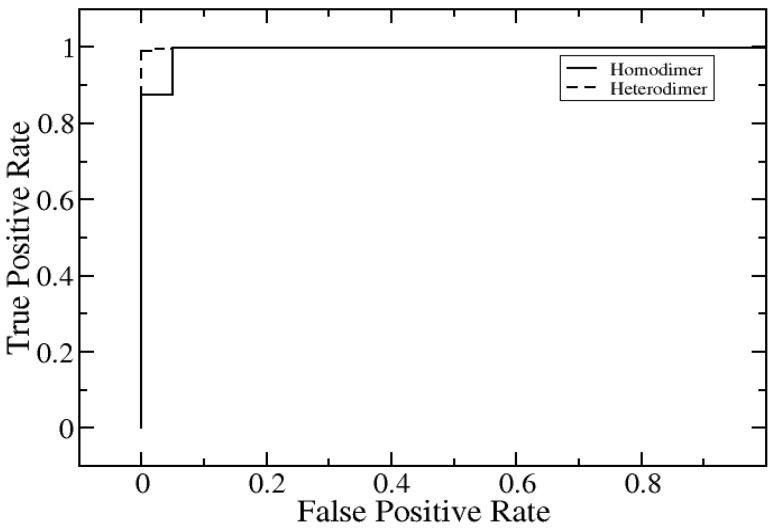
Prediction of disruptive and non-disruptive mutations for homodimer and heterodimer.

**Figure 8 ijms-21-02563-f008:**
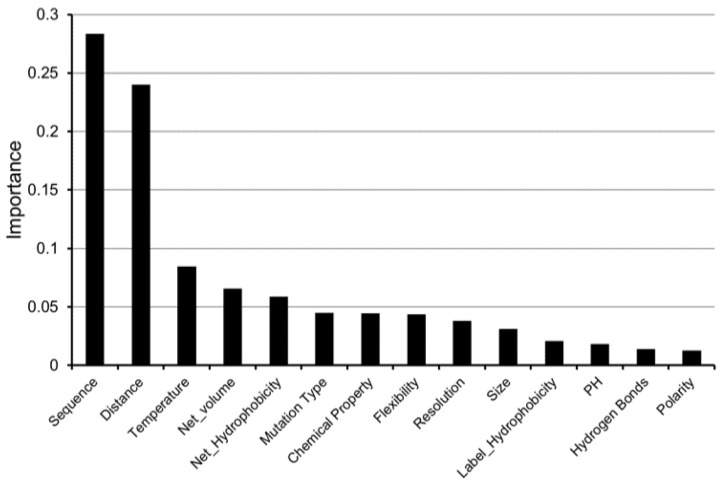
Importance level of each feature selected for SAAMBE-3D.

**Table 1 ijms-21-02563-t001:** Performance of SAAMBE-3D in predicting stabilizing, destabilizing, highly stabilizing and highly destabilizing mutations.

	Stabilizing and Destabilizing	Highly Stabilizing and Destabilizing
AUC	0.75	0.99
Sensitivity	0.82	0.95
Specificity	0.53	1
Precision	0.86	1
Accuracy	0.76	0.96
MCC	0.34	0.82

**Table 2 ijms-21-02563-t002:** Performance of SAAMBE-3D-DN in predicting disruptive and non-disruptive mutations for both homo- and hetero-dimers.

	Homo-Dimer	Hetero-Dimer
AUC	1	0.96
Sensitivity	0.99	1
Specificity	0.99	0.95
Precision	1	0.8
Accuracy	1	0.96

**Table 3 ijms-21-02563-t003:** Comparison of time of calculation for a single prediction between different ΔΔG predictors.

Method	Time of Calculation
mCSM-PPI2	42 seconds
SAAMBE-3D	0.21 seconds
MutaBind2	10 minutes
BindProfX	50 minutes
BeAtMuSiC	2 seconds

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
