# Peer review of "SAAMBE-3D: Predicting Effect of Mutations on Protein–Protein Interactions"

_ijms, 2020, doi:10.3390/ijms21072563_

Round 1
Reviewer 1 Report
The manuscript by Pahari et al., entitled “SAAMBE-3D: Predicting Effect of Mutations on Protein-Protein Interactions” present a novel software tool for calculating the impact of amino acid variations on formation of protein complexes. Overall, the manuscript is well-organized and of high quality, describing both the background and the results in details, while the language could profit of some improvements.
It is perhaps obvious to the authors (and other experts of the field) but it is unclear how the disruptive mutations in protein complexes, which were used for benchmarking the tool, were experimentally captured.
The authors state (p. 11, lines 436-438) that their algorithm “is an excellent too for early diagnostics”. It is unclear how it is projected and what diseases they think about. What would be an input data in case of a diagnostic application?
Being an experimental scientist, this reviewer is wondering whether the newly developed machine learning algorithm could be applied to predict and describe novel interactions, those frequently seen in protein interactomes?
Author Response
Thank you for your time and valuable comments.
Q1: It is perhaps obvious to the authors (and other experts of the field) but it is unclear how the disruptive mutations in protein complexes, which were used for benchmarking the tool, were experimentally captured.
Answer: To detect amino acid mutations that disrupt protein-protein interactions, a high-throughput mutagenesis and cloning platform was previously used to generate clones for each of the mutations reported in our testing set. These mutant clones were then transformed into yeast to perform yeast two-hybrid (Y2H) experiments in which mutations were scored as disruptive if they resulted in significantly reduced Y2H reporter activity relative to corresponding wild-type interactions. A statement is added on page number 9, line 446 to clarify this point.
Q2: The authors state (p. 11, lines 436-438) that their algorithm “is an excellent too for early diagnostics”. It is unclear how it is projected and what diseases they think about. What would be an input data in case of a diagnostic application?
Answer: Text is added on page 9, line 469 to address this question.
Q3: Being an experimental scientist, this reviewer is wondering whether the newly developed machine learning algorithm could be applied to predict and describe novel interactions, those frequently seen in protein interactomes?
Answer: yes, and this is demonstrated in our “blind tests” where the mutations were not used in the training and thus from point of view of the algorithm are “novel” mutations.
Reviewer 2 Report
The authors described a new method to effect of mutations on protein-protein interactions, named SAAMBE-3D. The method is a new development of the SAAMBE method [1], [2]. In the introduction, they described many different methods and explained well how they contribute to the field. The Methods are clear, thoroughly explained. The results are impressive and well illustrated. The method is compared to others with rigorous benchmarks and also on an independent dataset. The method is very fast and available as a stand-alone code and as a web server. Both, the server and the program worked and were very easy to install and use. I would encourage the authors to upload their code to http://github.com to promote the software and help others in giving feedback and reporting any issues. Minor: - Please put the software at the GitHub - At the web-sever, please add units of ddG (kcal/mol) - Change: when 80% of dataset-1 is used to train the model [Figure 2a]. -> when 80% of dataset-1 is used to train the model (Figure 2a) # and so on in other sentences, such as The results [Figure 5] indicate - Fix: MSE = 0.74kcal/mol -> MSE = 0.74 kcal/mol [1] M. Petukh, M. Li, and E. Alexov, “Predicting Binding Free Energy Change Caused by Point Mutations with Knowledge-Modified MM/PBSA Method,” PLoS Comput Biol, vol. 11, no. 7, p. e1004276, Jul. 2015. [2] M. Petukh, L. Dai, and E. Alexov, “SAAMBE: Webserver to Predict the Charge of Binding Free Energy Caused by Amino Acids Mutations.,” IJMS, vol. 17, no. 4, p. 547, Apr. 2016.
Author Response
Thank you for your time and valuable comments.
Q1: Please put the software at the GitHub
Answer: Done ; https://github.com/delphi001/SAAMBE-3D
Q2: At the web-sever, please add units of ddG (kcal/mol)
Answer: Done
Q3: Change: when 80% of dataset-1 is used to train the model [Figure 2a]. -> when 80% of dataset-1 is used to train the model (Figure 2a) # and so on in other sentences, such as The results [Figure 5] indicate
Answer: All corrections were made accordingly.
Q4: Fix: MSE = 0.74kcal/mol -> MSE = 0.74 kcal/mol
Answer: Corrected
Reviewer 3 Report
The manuscript presents a novel, machine learning-based method to predict the effect of mutations on protein-protein interactions in terms of the change in binding free energy, from the 3D structures of the complexes. The features are derived from both the sequences and the structures. The method is adequately tuned and cross-validated, and also tested in blind testing. The authors also perform a comparison with several existing methods. The methodology is sound, the description is very clear, and the testing and the analyses are thorough and convincing. The English of the manuscript is almost perfect. A small spelling error: "Concord" in line 84 should be Concoord.
I do not see any reason to request a revision, thus I recommend acceptance of the manuscript as it is (with the noted spelling error corrected, which can be done in the proofs).
Author Response
Thank you for your time and comments.
Q1: "Concord" in line 84 should be Concoord
Answer: Corrected
Reviewer 4 Report
In this paper, the authors reported a new method, SAAMBE-3D, for predicting the effects of mutations on protein-protein interactions. This method was based on SAAMBE, also developed by the authors. They improved the previous method by introducing a machine-learning based approach. The new method scored much higher in benchmark tests and calculation speed than some known methods. Thus, I recommend publishing this paper in this journal.
Author Response
Thank you for your time and positive comments